# Electrochemical Organophosphorus Pesticide Detection Using Nanostructured Gold-Modified Electrodes

**DOI:** 10.3390/s22249938

**Published:** 2022-12-16

**Authors:** Han-Wei Chang, Chien-Lin Chen, Yan-Hua Chen, Yu-Ming Chang, Feng-Jiin Liu, Yu-Chen Tsai

**Affiliations:** 1Department of Chemical Engineering, National United University, Miaoli 360302, Taiwan; 2Pesticide Analysis Center, National United University, Miaoli 360302, Taiwan; 3Department of Chemical Engineering, National Chung Hsing University, Taichung 40227, Taiwan

**Keywords:** nanostructured gold, methyl parathion, electrocatalytic activity, agricultural products

## Abstract

In this study, nanostructured gold was successfully prepared on a bare Au electrode using the electrochemical deposition method. Nanostructured gold provided more exposed active sites to facilitate the ion and electron transfer during the electrocatalytic reaction of organophosphorus pesticide (methyl parathion). The morphological and structural characterization of nanostructured gold was conducted using field-emission scanning electron microscopy (FESEM), X-ray photoelectron spectroscopy (XPS), and X-ray diffraction (XRD), which was further carried out to evaluate the electrocatalytic activity towards methyl parathion sensing. The electrochemical performance of nanostructured gold was investigated by electrochemical measurements (cyclic voltammetry (CV) and differential pulse voltammetry (DPV)). The proposed nanostructured gold-modified electrode exhibited prominent electrochemical methyl parathion sensing performance (including two linear concentration ranges from 0.01 to 0.5 ppm (R^2^ = 0.993) and from 0.5 to 4 ppm (R^2^ = 0.996), limit of detection of 5.9 ppb, excellent selectivity and stability), and excellent capability in determination of pesticide residue in real fruit and vegetable samples (bok choy and strawberry). The study demonstrated that the presented approach to fabricate a nanostructured gold-modified electrode could be practically applied to detect pesticide residue in agricultural products via integrating the electrochemical and gas chromatography coupled with mass spectrometry (GC/MS-MS) analysis.

## 1. Introduction

Extensive use of pesticides in agricultural products from agricultural areas results in increasing health risks and health costs, productivity loss, and degradation of the environment [1,2,3]. Significant pesticide residues are the most serious problem for food safety and sustainable agriculture. It stimulates the establishment of legal directives in food safety and continuous monitoring of pesticide residues in agricultural products. In Taiwan, in 2016, the Ministry of Health and Welfare proposed the “Five rings of food safety” policy to comprehensively protect consumers’ food supply. In 2017, the Council of Agriculture set the policy “Ten-year program for reducing chemical pesticide usage by half” to reduce the use of chemical pesticides to an optimal level, contributing to the development of sustainable agricultural practices. These policies have been devised to address the emerging food safety and sustainable agriculture challenges in Taiwan. A wide range of analytical methods for the determination of pesticide residues in agricultural products are based on gas chromatography coupled with mass spectrometry (GC/MS-MS) [4] and liquid chromatography-tandem with mass spectrometry (LC/MS-MS) [5]. Despite the excellent accuracy of these analytical instruments, the use of these instruments with more sensitive and selective detection means that rigorous sample clean-up procedures are required. Clearly, there exists a need for the development of fast and efficient methods to determine pesticide residues in agricultural products. Previous research provided a detailed list of the advantages and disadvantages for different analytical methods of pesticide residues [6,7,8]. It could be assessed that the electrochemical techniques might be an option for performing pesticide sensing due to the low cost, easy fabrication, high reliability, and good reproducibility [9,10,11,12,13,14]. Therefore, the integrated electrochemical sensing (rapid test method) and chromatographic/mass spectrometric-based instruments (precise and accurate analytical method) can open new possibilities in pesticide sensing for rapid screening and confirmatory testing of pesticide residues in agricultural products. 

Electrochemical pesticide sensing can be roughly classified into two types (i.e., enzymatic electrochemical pesticide sensing and non-enzymatic electrochemical pesticide sensing). Both types of electrochemical pesticide sensing have their own benefits and limitations. Many strategies have been formulated in improving the performance of enzymatic pesticide sensing through the acetylcholinesterase (AChE) inhibition mechanism. However, enzyme-based electrochemical sensing has some limitations, such as high fabrication cost, poor stability of enzymes, pH value dependent, and dedicated limitations [15,16,17]. Thus, the design of non-enzymatic electrochemical pesticide sensing is highly desirable. The performance of non-enzymatic electrochemical sensing depends greatly on the efficient electron transfer electrode substrates and excellent catalytic materials to improve sensitivity and selectivity. In the last decade, non-enzymatic electrochemical pesticide sensing based on noble metal nanomaterials has attracted tremendous scientific and technological interest due to its unique characteristics of facile preparation, multifunctional modifications, and unique catalytic/electrocatalytic properties, which make it attractive in the field of catalysis, energy storage, biomedicine, and electrochemical sensing [18,19,20]. 

Anandhakumar et al. reported that atomic gold was successfully structured on the surface of a gold electrode using the electrodeposition method for constructing non-enzymatic electrochemical methyl parathion sensing. Atomic gold structured on a gold electrode possessed high surface to volume ratio, large binding sites, electronic effect, and a mixed diffusion regime to enable further improvements in the performance of electrochemical sensing for determining methyl parathion [18]. Junior et al. also reported that sensitive and selective non-enzymatic electrochemical methyl parathion sensing at the fabrication of nanoporous gold on a gold electrode was prepared successfully by using anodization followed by the electrochemical reduction method. The excellent pesticide sensing performance could be ascribed to enhancing the electroactive surface area of nanostructured electrodes that increased the number of catalytic sites and crystalline facets in this nanoporous gold film structure to maximize its catalytic performance for further application to non-enzymatic methyl parathion sensing [21]. As reported in previous studies on the mechanism of non-enzymatic electrochemical methyl parathion sensing, methyl parathion was initially electrochemically irreversibly reduced with a cathodic scan following the direct electron transferring process (4e^−^ and 4H^+^) from the conversion of nitro (–NO_2_) on methyl parathion to hydroxylamine group (−NH–OH). Subsequently, the hydroxylamine group (−NH–OH) on methyl parathion could be reversibly oxidized and reduced with the nitroso group (–NO) by the reversible transformation of 2e^−^ and 2H^+^ [21,22,23].

In this work, our efforts aimed at the development of the simultaneous rapid screening (the electrochemical analysis) and confirmatory testing (the gas chromatography coupled with mass spectrometry (GC/MS-MS) analysis) of pesticide residues on agricultural products. The nanostructured gold-modified electrode was successfully fabricated by chronoamperometric electrodeposition. The as-prepared nanostructured gold-modified electrode, as an advanced electrocatalyst, ensured highly exposed active sites and efficient charge transfer across the interface, which was favorable for electrocatalytic activity enhancement toward pesticide sensing and improved the non-enzymatic electrochemical pesticide sensing performance. The proposed nanostructured gold-modified electrode might enable more opportunities for the electrochemical determination of methyl parathion in real fruit and vegetable samples (bok choy and strawberry), verified through gas chromatography–mass spectrometry. It presents a good development in the rapid screening and confirmatory testing of pesticide residues in agricultural products. 

## 2. Materials and Methods

### 2.1. Reagents

Hydrogen tetrachloroaurate (III) trihydrate (HAuCl_4_·3H_2_O), p-nitrophenol, and p-aminophenol were obtained from Alfa Aesar (Ward Hill, MA, USA). Methyl parathion (MP), potassium chloride, sodium perchlorate, sodium sulfate, and sodium chloride were purchased from Sigma-Aldrich (St. Louis, MO, USA). Stock solution of methyl parathion (1000 ppm) was prepared in 20% acetonitrile and 0.1 M phosphate-buffered saline (PBS) for further experiments. Disodium hydrogenphosphate (Na_2_HPO_4_) and sodium dihydrogenphosphate anhydrous (NaH_2_PO_4_) were purchased from Showa Chemical Co., Ltd. (Tokyo, Japan). The deionized water (DI water) was produced by Milli-Q water purification system (Millipore, MA, USA). All chemicals were analytical grade and were used as received without further purification.

### 2.2. Preparation of Nanostructured Gold-Modified Electrode

Prior to each modification, bare Au electrode (2 mm diameter, CH Instrument, Bee Cave, TX, USA) was carefully polished with alumina powders (0.3 and 0.05 μm) on a polishing cloth and rinsed thoroughly with deionized water, sonicated in ethanol and dried at ambient temperature. In this work, electrodeposition of nanostructured gold was carried out in an electrochemical cell for mounting the bare Au electrode using a two-electrode system at constant potential (the bare Au electrode as the working electrode and Pt as the counter electrode). Subsequently, the nanostructured gold was electrodeposited by chronoamperometry (CA) in a 0.5 M H_2_SO_4_ solution containing 1 mM HAuCl_4_·3H_2_O at an applied potential of 2.5 V with various electrodeposition times (0, 15, 30, 60, 80, and 100 s) and the obtained electrode was named as nanostructured gold-modified electrode. After that, the obtained nanostructured gold-modified electrode was washed with ethanol and DI water three times, dried in an oven at 60 °C, and collected for subsequent treatment. 

### 2.3. Preparation of Real Vegetable and Fruit Samples with GC–MS/MS

The detections of MP in bok choy and strawberry bought from a local supermarket were carried out to evaluate the practical application and quantified using both gas chromatography tandem mass spectrometry (GC-MS/MS) and electrochemical detection. For the sample preparation, bok choy and strawberry were prepared using the Quick, Easy, Cheap, Effective, Rugged, and Safe (QuEChERS) extraction method. The extraction and cleanup steps in the QuEChERS method combined with a GC–MS/MS and electrochemistry for the quantification MP are summarized in Appendix A. 

### 2.4. Apparatus

The morphology was characterized using field-emission scanning electron microscopy (FESEM, JSM-7410F, JEOL, Akishima, Japan). The chemical structure and composition were characterized using X-ray photoelectron spectroscopy (XPS, PHI-5000 Versaprobe, ULVAC-PHI, Chigasaki, Japan). The structure of crystalline phases was characterized by X-ray diffraction (XRD) using a D8 Discover (Bruker, Darmstadt, Germany) X-ray diffractometer with Cu Kα radiation (λ = 0.154 nm) in the 2θ range from 10° to 90°. Electrochemical measurements were performed using a three-electrode system comprising the obtained nanostructured gold-modified electrode as working electrode, a platinum wire as counter electrode, and a Ag/AgCl (3 M KCl) as reference electrode via an electrochemical analyzer (Autolab, model PGSTAT30, Eco Chemie, Utrecht, The Netherlands). All electrochemical measurements were conducted in a solution composed of the 20% acetonitrile and 0.1 M phosphate-buffered saline (PBS) as supporting electrolyte in the absence and presence of methyl parathion (MP) at ambient temperature. Degradation products of MP were characterized by using GC-MS/MS, composed of gas chromatography (GC) (GC 2030, Shimadzu Co., Ltd., Kyoto, Japan) equipped with an AOC-6000 auto-sampler using a 10 μL syringe (the injection volume was set at 1 μL and a flow of helium at 1 mL/min was used as the carrier gas) and hyphenated to a triple-quadruple tandem mass spectrometer (MS) (TQ8040 NX, Shimadzu Co., Ltd., Kyoto, Japan). The capillary column was DB-5ms UI (0.25 μm film thickness, 0.25 mm ID and 30 m length) (Agilent Technologies, Santa Clara, CA, USA). The column oven temperature was increased from 60 °C (1 min hold time) to 170 °C at a rate of 40 °C/min and then to 220 °C at a rate of 10 °C/min to 320 °C (3 min hold time) at a rate of 5 °C/min. The temperatures of the injector and ion source were 280 °C and 300 °C, respectively. The triple-quadrupole MS was applied in the multiple reaction monitoring (MRM) with electron impact ionization (EI) at 70 eV.

## 3. Results

The morphologies of the bare Au electrode and nanostructured gold-modified electrode were characterized using FESEM. Figure 1a,b show the FESEM images of the bare Au electrode and nanostructured gold-modified electrode. In contrast to the bare Au electrode, they confirm that spherical Au nanoparticles are successfully electrodeposited on the bare gold electrode surface and the average diameter of the as-prepared gold nanoparticles is in a few tens of nanometers. They show that smaller-size spherical Au nanoparticles on the bare gold electrode surface ensure the full exposure of electrocatalytically active sites and further facilitate the charge transfer at the interface between the electrode and the electrolyte, contributing to the enhanced intrinsic interfacial charge transfer and charge transport kinetics under an electrochemical condition. It is beneficial to boost the comprehensive electrochemical performance of potential sensing applications.

The structure and phase composition of the nanostructured gold-modified electrode were characterized using XRD pattern in comparison with the standard patterns of Au (JCPDS No. 04-0784) [24], as shown in Figure 2. The XRD result of the nanostructured gold-modified electrode revealed typical Au reflection peaks at 2θ = 38.3°, 44.5°, 64.6°, and 77.6°, which corresponded to the (111), (200), (220), and (311) lattice planes of Au. The average crystallite/particle size of deposited Au on the nanostructured gold-modified electrode was then estimated from the Debye-Scherrer equation in the XRD data [25].

Equation (1) shows the Debye-Scherrer equation
D = Kλ/βcosθ(1)
where D was the average crystallite/particle size (nm), K was the Scherrer constant (0.9), λ was the X-ray wavelength (0.154 nm), β was the full width at half-maximum intensity (FWHM) of the diffraction peak, and θ was the diffraction angle. The estimated average crystallite/particle size of deposited Au on the bare Au electrode was 24.12 nm, which was in good agreement with the size obtained from the FESEM image. 

The surface elemental composition and valance states of the nanostructured gold-modified electrode were characterized using XPS, as shown in Figure 3. The Au 4f, C 1 s, Au 4d, and O 1s elements could be identified in the full survey spectra for the nanostructured gold-modified electrode (Figure 3a), which confirmed the attachment of Au nanoparticles on the surface of the bare Au electrode. In order to further characterize the chemical states of the Au element on the nanostructured gold-modified electrode, it was characterized in detail by high-resolution Au 4f XPS spectrum. In the high-resolution XPS spectrum of the Au 4f region (Figure 3b), the nanostructured gold-modified electrode exhibited the Au 4f split peaks at 84.0 eV and 87.7 eV, accounting for Au 4f_7/2_ and Au 4f_5/2_, respectively, thus, confirming the existence of metallic Au nanoparticles on the nanostructured gold-modified electrode [26].

The electrochemical characteristics of the bare Au electrode and nanostructured gold-modified electrode were performed by cyclic voltammetry (CV) and differential pulse voltammetry (DPV). Figure 4 shows the CV curves of the bare Au electrode and nanostructured gold-modified electrode in the 20% acetonitrile and 0.1 M phosphate-buffered saline (PBS) in the absence (dashed lines) and presence (solid lines) of MP at a scan rate of 200 mV s^−1^. In the absence of MP, both electrons show a significant reduction peak located at about −0.1 V, which should be ascribed to the reduction in the residual oxygen in the electrolyte solution. In order to avoid the effects of residual oxygen in the further electrochemical experiments, the removal of dissolved oxygen from the electrolyte solution by purging the solution with nitrogen for 20 min ensured no residual dissolved oxygen within the electrolyte solution [27]. In Figure 4, there is no well-defined peak and redox peak at the bare Au electrode in the presence of MP. However, an irreversible reduction peak and a pair of well-defined reversible redox peaks appear in the CV curve of the nanostructured gold-modified electrode, suggesting that the electrodeposited nanostructured gold on the bare Au electrode is an excellent electrocatalytic electrode material for nonenzymatic electrochemical detection of MP. The irreversible peak at about −0.2 V is attributed to the reduction of the nitrophenyl group (−NO_2_) on MP capturing four electrons to form a hydroxylamine group (−NH–OH), and the reversible peaks at about 0.1 V are related to the redox reaction between the hydroxylamine group and nitroso group (–NO) on MP involving a two-electron process redox reaction [28], demonstrating that the nanostructured gold-modified electrode can be employed for the qualitative and quantitative detection of MP. The reaction mechanism for nonenzymatic electrochemical detection of MP could be expressed as Figure 1, which enhanced the understanding of the reaction mechanism for nonenzymatic electrochemical detection of MP.

To further elucidate the electrochemical behavior of nanostructured gold modified on the Au electrode, we measured electroactive surface areas using the Randles–Sevcik equation. Figure 5 displays the CV curves of the bare Au electrode (Figure 5a) and nanostructured gold-modified electrode (Figure 5b) in ferricyanide solution (1 mM) containing 0.1 M KCl at different scan rates from 20 to 200 mV s^−1^. It is shown that the peak current increases with increasing scan rate, with anodic peak current (I_pa_) and cathodic peak current (I_pc_) increasing linearly with the square root of the scan rate (υ^1/2^), see Figure 5c,d. The linear regression equations for the bare Au electrode and nanostructured gold-modified electrode could be expressed as I_pa_ (I_pc_) (μA) = −0.85 (−2.16) + 0.73 (−0.69) υ^1/2^ ((mV s^−1^)^1/2^) (R^2^ = 0.999 (0.999)) (bare Au electrode) and I_pa_ (I_pc_) (μA) = −1.81 (−1.62) + 1.01 (−0.91) υ^1/2^((mV s^−1^)^1/2^) (R^2^ = 0.999 (0.999)) (nanostructured gold-modified electrode), respectively. According to the following Randles–Sevcik equation [29], I_pa_(I_pc_) = 2.69 × 10^5^ n^3/2^AD^1/2^C_0_υ^1/2^, D is the diffusion coefficient of ferricyanide (D = 7.6 × 10^−6^ cm^2^ s^−1^ for ferricyanide), I_pa_ (I_pc_) is the anodic (cathodic) peak current (A), C_0_ is the concentration of the ferricyanide (10^−6^ mol cm^−3^), A is the electroactive area (cm^2^), n is the number of transferred electrons, and υ^1/2^ is the square root of scan rate ((V s^−1^)^1/2^). The calculated electroactive area is 0.030 and 0.042 cm^2^ for the bare Au electrode and nanostructured gold-modified electrode, indicating that the nanostructured gold-modified electrode has relatively high electrocatalytic active sites compared with the bare Au electrode, which would greatly enhance the electrochemical sensing performance.

To confirm the reaction mechanism of MP involving the irreversible reduction reaction from the nitrophenyl group (−NO_2_) on MP capturing four electrons to form the hydroxylamine group (−NH–OH), the Laviron equation given for the irreversible electrode process was used. Figure 6a displays the CV curves of the nanostructured gold-modified electrode in the 20% acetonitrile and 0.1 M PBS in the presence of 4 ppm MP at different scan rates from 50 to 400 mV s^−1^. It is shown that a negative shift in the irreversible cathodic peak potential with increasing scan rates is obtained (Figure 6a). Figure 6b shows that a plot of logarithm of the scan rate (lnν) vs. the irreversible cathodic peak potential is linear, and the linear regression equations could be expressed as E_p_ (V) = −0.493 − 0.038 lnν(V s^−1^) (R^2^ = 0.927). According to the following Laviron equation [30], E_p_ = E^0^ + (RT/αnF)ln(RTk^0^/αnF) + (RT/αnF)lnυ, E_p_ is the peak potential (V), E^0^ is the formal peak potential (V), R is the universal gas constant (8.314 J mol^−1^ K^−1^), T is the temperature (298 K), α is the charge transfer coefficient, n is the amount of electron transfer, F is the Faraday constant (96,500 C mol^−1^), k^0^ is the standard heterogeneous electron transfer rate constant, and υ is the scan rate (V s^−1^). The calculated values of αn and n further confirmed that the reaction mechanism of MP for the nanostructured gold-modified electrode involving the irreversible reduction undergoes a four-electron transfer process in electrochemical MP sensing.

In order to achieve an optimal electrochemical MP sensing performance, the pH of the electrolyte solution, the adsorption time of the pesticide, and the electrodeposition time of the nanostructured gold were examined to gain more insight into the optimization of the operating parameters affecting the electrochemical MP sensing performance. To evaluate the impact of pH on electrolyte solution, the pH impact on the peak current response of the nanostructured gold-modified electrode to MP with pH adjustment in the 20% acetonitrile and 0.1 M phosphate-buffered saline (PBS) as electrolyte solution (in a range of 3.0~8.5) was performed by DPV (Figure 7a). The optimal parameters for DPV-based experiments contained increment potential = 0.004 V, amplitude = 0.04 V, pulse period = 0.2 s, and pulse width = 0.05 s. It could be seen that the peak current response increased with increasing pH value to 4.0 and then decreased as pH increased further. The maximum peak current response was obtained in electrolyte solution pH 4.0. This result was caused by the degradation of MP in basic electrolyte solutions and H^+^ could be involved in the irreversible reduction of MP [31]. The impact of the MP adsorption time on the nanostructured gold-modified electrode was performed by recording the DPV peak current response of MP at different adsorption time (Figure 7b). The peak current response increased gradually with increasing adsorption time to 60 s and then reached a relative plateau for adsorption time higher than 60 s, indicating that the adsorption of MP on the nanostructured gold-modified electrode reached saturation [31]. Figure 7c shows the nanostructured gold-modified electrode with different electrodeposition time being characterized using DPV in the presence of MP. The impact of electrodeposition time was examined by controlling the electrodeposition time using chronoamperometry (CA) in a 0.5 M H_2_SO_4_ solution containing 1 mM HAuCl_4_·3H_2_O at an applied potential of 2.5 V (from 0 to 100 s). Figure 7c shows that the peak current response gradually increases with the increase in the electrodeposition time of the nano-Au film. When the electrodeposition time exceeds 60 s, the peak current response reaches the maximum and remains almost unchanged. However, by continuously increasing the deposition time beyond 60 s, there was a slight decrease in the response current, which could be due to the limited increase in electrochemically active surface area to increase mass transfer limitation [32]. To obtain an excellent sensing performance in MP detection, 4.0 pH of electrolyte solution, 60 s adsorption time of MP, and 60 s electrodeposition time of nanostructured gold were selected as the optimal operating conditions and adopted in the following experiments.

Figure 8 displays the DPV peak current response of the nanostructured gold-modified electrode in 20% acetonitrile and 0.1 M PBS with successive addition of various MP concentrations (0~4 ppm) to estimate the applicability of the fabricated electrochemical sensing. Figure 8a shows that the nanostructured gold-modified electrode exhibits a better DPV peak current response towards MP sensing. It can be seen that, with increasing MP concentration, the DPV peak current response increases gradually. Furthermore, the DPV peak in cathodic scan shifts towards negative potential with increasing MP concentration, which is attributed to the diffusion limitation of MP from electrolyte to electrode surface in the working electrochemical conditions [33]. The DPV peak current response in an MP concentration range between 0 and 4 ppm was collected to plot the corresponding calibration curve, illustrated in Figure 8b. The plot of the DPV peak current response versus the increasing MP concentration exhibits two linear relationships with the MP concentration in ranges from 0.01 to 0.5 ppm (R^2^ = 0.993) and from 0.5 to 4 ppm (R^2^ = 0.996), with sensitivity of 10.7 μA ppm^−1^ cm^−2^. The corresponding detection limit of MP is 5.9 ppb is based on 3 S_b_/m (S_b_ is the standard deviation of the blank signals for n = 3 and m is the slope of the calibration curve). The determined linear range and detection limit values for the nanostructured gold-modified electrode are comparable to other previously reported electrochemical MP sensing based on gold nanomaterials [31,34,35,36,37] (Table 1). 

The selectivity is a key factor to evaluate the sensing capacity of the electrode in practical applications. To demonstrate the selectivity of the nanostructured gold-modified electrode for MP sensing, the effect of possible inorganic ions (such as Na_2_SO_4_, KCl, NaCl, NaClO_4_, NaNO_3_) and electroactive nitrophenyl derivatives (such as p-nitrophenol and p-aminophenol) on the analytical response of the designed electrode was examined. In Figure 9, the DPV peak current response of the nanostructured gold-modified electrode displays an acceptable interference effect via the addition of possible interfering species (with concentrations 10-fold higher than the MP) in the presence of 1 ppm MP. The calculated relative standard deviation (RSD) is 7.36%. Such results still achieved an accepted target value of RSD because most existing pesticide residue analytical methods with the values of RSD < 20% [38] illustrated that there were many challenges remaining in the field of pesticide residue analysis. This could further stimulate researchers to promote the continued transformation of scientific and technological achievements on the analysis of pesticide residues in agricultural products.

Subsequently, to further evaluate the stability of the nanostructured gold-modified electrode, the repeatability (Figure 10a) and reproducibility (Figure 10b) of the electrode were examined to assess the efficiency of the electrode by recording the peak current response of 4 ppm MP in the 20% acetonitrile and 0.1 M PBS. The repeatability was examined using seven consecutive measurements. The relative standard deviation (RSD) of the nanostructured gold-modified electrode was 4.73%. The reproducibility was examined utilizing six different electrodes prepared with the same method and the RSD obtained was 6.16%, demonstrating that this proposed electrode presented satisfactory stability and could be used as a realistic feasible electrode material for MP sensing. 

Real sample analysis is also explicitly considered as a key factor to evaluate the feasibility of the proposed sensor for practical applications. Real sample detection for the nanostructured gold-modified electrode was performed to determine the MP concentration, with the successive addition of various MP concentrations (0~8 ppm) in real samples (bok choy and strawberry) (Figure 11). For the bok choy sample, the nanostructured gold-modified electrode exhibited two linear ranges from 0.1 to 2 ppm (R^2^ = 0.995) and from 2 to 8 ppm (R^2^ = 0.993), with sensitivity of 1.05 μA ppm^−1^ cm^−2^ and an LOD of 0.1 ppm. For strawberry, it exhibited two linear ranges from 0.1 to 2 ppm (R^2^ = 0.992) and from 0.5 to 4 ppm (R^2^ = 0.997), with sensitivity of 1.97 μA ppm^−1^ cm^−2^ and an LOD of 0.05 ppm. Moreover, the sensitivity obtained in real samples was lower than that obtained in phosphate-buffered saline, which was probably due to the matrix effect [39]. Then, the feasibility of the proposed nanostructured gold-modified electrode with the electrochemical and gas chromatography coupled with mass spectrometry (GC/MS-MS) analysis was separately demonstrated for the bok choy and strawberry samples. The obtained concentration values at the proposed nanostructured gold-modified electrode using electrochemical analysis were compared with GC-MS analysis. The results for real sample analyses (three repeated measurements) indicated great recovery and repeatability and were in good agreement with the results obtained by the reference method (GC/MS-MS). The comparison results are given in Table 2. Therefore, these results indicate that the nanostructured gold-modified electrodes have accurate and reliable detection results for electrochemical pesticide sensing for their practical application.

## 4. Conclusions

In this study, the electrodeposited nanostructured gold exposed more active sites to enrich methyl parathion molecules around the interface of electrolyte/electrode and further improved the electrocatalytic performance towards methyl parathion sensing. The proposed nanostructured gold-modified electrode exhibited prominent electrocatalytic activity towards methyl parathion sensing (including two linear concentration ranges from 0.01 to 0.5 ppm (R^2^ = 0.993) and from 0.5 to 4 ppm (R^2^ = 0.996), limit of detection of 5.9 ppb, and excellent selectivity), demonstrating their applicability in real samples (bok choy and strawberry). The obtained results from the proposed electrode are consistent with those from GC/MS-MS analysis, manifesting its promising potential in practical application. Therefore, the integrated electrochemical sensing (rapid test method) and chromatographic/mass spectrometric-based instruments (precise and accurate analytical method) open new possibilities in pesticide sensing for rapid screening and confirmatory testing of pesticide residues in agricultural products. 

## Data Availability

Not applicable.

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
