# Peer review of "Electrochemical Organophosphorus Pesticide Detection Using Nanostructured Gold-Modified Electrodes"

_sensors, 2022, doi:10.3390/s22249938_

Round 1
Reviewer 1 Report
The article on the detection of organophosphorus pesticides by electrochemical sensing carries significant scientific value. The authors claimed that prepared nano-au film to modify au-based electrodes significantly increases the effective active sides and was successfully validated by several experimental procedures. However, authors can show (if possible) atomic force microscopy-based surface roughness measurement will definitely enhance its scientific values.
In the introduction section, citations seem slightly misarranged. Authors are recommended to rearrange the citations. For example, lines no 36-40, 43-47 and 73-74 should consider for the arrangements.
Overall the article is sound for reading. It can be accepted with minor corrections.
Reviewer 2 Report
The manuscript by Han-Wei et al. investigated an method for the detection of organophosphorus pesticide depending on nano-Au film modified electrode. The scientific quality of the study meets the journal standard and the manuscript has been well-written following the journal guideline. The manuscript could be published after some minor corrections
Title: Too long. Please consider to make it short and direct
Abstract: Missing numerical value
Line 28: what do you mean real samples?
Keywords: Use different keywords rather than title
Line 43: Rewrite the sentence and add a reference
Line 52: Please add reference
Line 73: Wrong citation, Year/citation incomplete
Line 88-95: Research objective need to write properly. What is the hypothesis
Line 121: What do you mean by real samples? Please explain for the general reader.
Figure 1 could go in the supplementary and you could adjust the figure number.
Figure 8: Are you missing stat letter or they are not significantly different. Please write clearly in the figure footnote.
Figure 9: Please add stat letter and show whether they are significantly different or not
Reviewer 3 Report
In this study, nano-Au film was electrochemically deposited on the bare Au electrode for detection of organophosphorus pesticide (methyl parathion). The results obtained using the electrochemical and gas chromatography coupled with mass spectrometry (GC/MS-MS) analysis clearly demonstrated that the presented approach to fabricate nano-Au film modified electrode can be applied for practical application to detect pesticide residue in real samples. Although being interesting, I find that there are some major issues with the paper that require addressing prior to this being considered for publication in this journal. I have identified the main points for consideration below:
1. This manuscript has some spelling typos, style errors and grammatical errors. Pleases carefully check the whole manuscript.
2. The novelty of this study is very poor, because the sensing materials were often used for detection of organophosphorus pesticide.
3. The sensing mechanism of the proposed sensor should be clearly illustrated in a scheme and discussed combining some electrochemical experimental results.
4. The EIS plots of modified and bare Au electrode should be added.
5. The authors should compare the electrochemical response of MP on the bare Au electrode and Au film modified Au electrode.
6. x-axis in Figure 7a is wrong, please correct it. In addition, why did the peak potential negatively shift with MP concentration increasing.
7. The sensitivity of the sensor in buffer and real samples such as bok choy and strawberry should be compared.
8. The repeatability, producibility and stability of the proposed sensor should be added in the revised manuscript.
9. The advantages of electrochemical sensors should be added in the introduction section. Some recent related references are also recommended to be cited, such as Journal of Hazardous Materials 436 (2022) 129107; Materials Today Chemistry 26 (2022) 101043; Microchemical Journal 179 (2022) 107515; TrAC Trends in Analytical Chemistry, 2022, 146, 116487.
Reviewer 4 Report
Dear Editor;
This manuscript by Chang et al. explains a new sensor as nano-Au film-modified electrode for organophosphorus pesticide (methyl parathion). It is seen that the manuscript was written in a hurry and some important points were overlooked. Therefore, a major revision of the following points is required:
1. English of the article must be improved throughout the manuscript.
2. Existing literature on methyl parathion (MP) determinations should be included in detail.
3. The advantages of electrochemical methods should be emphasized in the Introduction. The electrochemical advantages of 2 or 3 sentences should be added. The following will be a study guide.
Ø https://doi.org/10.1080/03067319.2014.940340
Ø https://doi.org/10.1016/j.jelechem.2021.115389
4. How much more sensitive is the modified electrode than the bare electrode? In addition, the catalytic properties of the electrodes should be emphasized.
5. In Figure 7, the anodic peak shifted to more negative regions due to the increasing concentration of methyl parathion (MP). The reason for this needs to be explained.
6. Table 1 and Table 2 should be rearranged to afford significant figures.
7. Electrode behavior of methyl parathion (MP) should be examined in detail with cyclic voltammetry. In particular, information should be given about the material transport processes. https://doi.org/10.1007/s11694-020-00457-6
What is the Ep(V)= logv(V/s) and Ep(V)=logv(V/s) ?
8. What is the substance (MP): interference ratio in the interference effect study? In addition, it is seen that the interference rate of inactive substances such as NaNO3 and Na2SO4 is more than 10%. The reason for this needs to be explained.
9. The voltammograms in Figure 9 show shifts in the potentials of the peaks. What is the reason for this?
10. Anodic electrode mechanism of MP can also be included in the article.
Round 2
Reviewer 3 Report
The authors have addressed the comments. There is no further comment.
Author Response
We thank the Reviewer for the valuable suggestion that helped us improve this manuscript.
Reviewer 4 Report
The article is at an acceptable level in this new state.
Author Response

(The authors gave the same response as above.)
